# Papillary Thyroid Carcinoma Tissue miR-146b, -21, -221, -222, -181b Expression in Relation with Clinicopathological Features

**DOI:** 10.3390/diagnostics11030418

**Published:** 2021-03-02

**Authors:** Aistė Kondrotienė, Albertas Daukša, Daina Pamedytytė, Mintautė Kazokaitė, Aurelija Žvirblienė, Dalia Daukšienė, Vaida Simanavičienė, Raimonda Klimaitė, Ieva Golubickaitė, Rytis Stakaitis, Valdas Šarauskas, Rasa Verkauskienė, Birutė Žilaitienė

**Affiliations:** 1Institute of Endocrinology, Medical Academy, Lithuanian University of Health Sciences, LT-50161 Kaunas, Lithuania; aiste.kondrotiene@lsmuni.lt (A.K.); mintaute.kazokaite@lsmuni.lt (M.K.); dalia.dauksiene@lsmu.lt (D.D.); raimonda.klimaite@lsmuni.lt (R.K.); rasa.verkauskiene@lsmuni.lt (R.V.); 2Institute of Digestive Research, Medical Academy, Faculty of Medicine, Lithuanian University of Health Sciences, LT-50161 Kaunas, Lithuania; albertas.dauksa@lsmuni.lt; 3Institute of Biotechnology, Life Sciences Center, Vilnius University, LT-10257 Vilnius, Lithuania; daina.pamedytyte@gmc.vu.lt (D.P.); aurelija.zvirbliene@bti.vu.lt (A.Ž.); vaida.simanav@gmail.com (V.S.); 4Department of Genetics and Molecular Medicine, Lithuanian University of Health Sciences, LT-50161 Kaunas, Lithuania; ieva.golubickaite@lsmuni.lt; 5Laboratory of Molecular Neurooncology, Neuroscience Institute, Lithuanian University of Health Sciences, LT-50161 Kaunas, Lithuania; rytis.stakaitis@lsmuni.lt; 6Department of Pathology, Lithuanian University of Health Sciences, LT-50161 Kaunas, Lithuania; valdas.sarauskas@lsmu.lt

**Keywords:** papillary thyroid carcinoma, miR-146, miR-21, miR-221, miR-222, miR-181b, clinicopathologic features, overall survival

## Abstract

We analyzed miR-146b, miR-21, miR-221, miR-21, and miR-181b in formalin fixed paraffin-embedded papillary thyroid carcinoma (PTC) tissue samples of 312 individuals and evaluated their expression relationship with clinicopathological parameters. A higher expression of miR-21 was related to unifocal lesions (*p* < 0.011) and autoimmune thyroiditis (0.007). miR-221, miR-222 expression was higher in the PTC tissue samples with extrathyroidal extension (*p* = 0.049, 0.003, respectively). In a group of PTC patients with pT1a and pT1b sized tumors, the expression of miR-146b, miR-21, miR-221, and miR-222 in PTC tissue samples was lower than in patients with pT2, pT3, and pT4 (*p* = 0.032; 0.0044; 0.003; 0.001; 0.001, respectively). Patients with lymph node metastases had higher expression of miR-21, -221, -222, and -181b (*p* < 0.05). A high expression of miR-146b, miR-21, miR-221 panel was associated with decreased overall survival (OS) (Log rank *p* = 0.019). Univariate analysis revealed that presence of metastatic lymph nodes and high expression of miR-146b, miR-21, and miR-221 panels were associated with increased hazard of shorter OS. After multivariate analysis, only sex (male) and age (≥55 years) emerged as independent prognostic factors associated with shorter OS (HR 0.28 (95% CI 0.09–0.86) and HR 0.05 (95% CI 0.01–0.22), respectively). In conclusion, 5 analyzed miRs expression have significant relations to clinicopathologic parameters so further investigations of these molecules are expedient while searching for prognostic PTC biomarkers.

## 1. Introduction

Papillary thyroid carcinoma (PTC) is the most common endocrine malignancy, which makes up to 80–85% of thyroid cancer [1]. According to the European Network of Cancer Registries, in Europe an estimated 53,000 thyroid cancer cases were newly diagnosed in 2012 [2]. In the same year, 6300 Europeans were estimated to have died of thyroid cancer [2]. In particular, the highest estimated age-standardized incidence ratio in Europe is recorded in Lithuania (15.5 cases per 100,000 person-years) [2]. The ten-year survival rate is usually higher than 80–90% for a patient with PTC after indicated treatment [2,3]. However, local regional recurrences occur in up to 30% of patients with PTC [4]. The current problem is in properly identifying and evaluating a patient’s prognosis, applying an adequate treatment and a life-long screening plan. Inadequately managed patients result in potentially higher fatal outcomes due to the lack of sufficient prognostic data/markers, inadequate periodic individualized follow-up risk assessments, and/or insufficient initial treatment [5].

Prognostic risk factors for PTC recurrence are: ≥55 years age, large tumor size, extra thyroidal extension, lymph node metastasis, distant metastasis, and lymphovascular invasion, but even these are not able to predict prognostic recurrence in many cases or not until the tumor has reached a more advanced stage [6,7]. Additionally, half of distant metastases are only diagnosed during the follow-up period [8]. Over the last decades, an increasing incidence of thyroid cancer has been observed [9], mostly due to an increased sensitivity of diagnostic and follow-up investigations. These cases are generally attributable to newly diagnosed tumors that are tiny, localized, and asymptomatic (usually microcarcinomas) but require lifelong follow-up care. [5]. It is still controversial how much such patients will benefit from additional therapy and monitoring.

Recently, attempts were made to revealing molecular markers of thyroid cancer. The European Thyroid Association advises investigating BRAF, RET/PTC, PAX8/PPARG, RAS mutations, TERT promoter, and TP53 in case of hesitation about further clinical action depending on Bethesda classes, but these biomarkers are not recommended for follow-up yet [10]. As miR molecules demonstrate a potential value in PTC diagnostics and follow up, further miR investigations are encouraged in the same guidelines [10].

miRs are small, highly conserved non-coding RNA molecules involved in the regulation of gene expression. [8,11]. The majority of miRs are transcribed from DNA sequences into primary miRs and processed into precursor miRs, and finally mature miRs. In most cases, miRs interact with the 3′ untranslated region (3′ UTR) of target mRNAs to induce mRNA degradation and translational repression [12]. However, the interaction of miRs with other regions, including the 5′ UTR, coding sequence, and gene promoters, have also been reported [12]. Under certain conditions, miRs can also activate translation or regulate transcription [12]. Recently, numerous studies have shown that miRs are misexpressed in tumors when compared to normal tissue [13]. miR-222, -221, -21, -146b, and -181b are the molecules that showed potential as biomarkers in PTC tissue studies already [14,15,16,17,18,19,20,21,22]. Moreover, previous studies of PTC patients have revealed enhanced expression of certain miRs in recurrent PTC compared to non-recurrent [17,19,21,22,23,24,25].

miR expression levels’ relation to clinicopathological features of PTC are being analyzed in studies as well. For example, miR-146b, -221-222 tend to be upregulated in PTC tissue samples with extra thyroidal invasion [15,24,25,26,27]. Moreover, studies comparing PTC patients with and without lymph node metastases showed upregulated miRs (miR-146, miR-221, miR-222, and miR-146a) in patients with present lymph node metastases [15,18,25,28]. Furthermore, miRs (miR-146b and miR-222) which both displayed statistical significance (*p* < 0.05) had lower expression in TNM stage I/II against III/IV stage patients [15,26]. However, the data is still limited.

The aim of our study was to analyze expression of five miR molecules (miR-146b, miR -21, miR -221, miR -222, and miR -181b) in PTC formalin-fixed, paraffin-embedded (FFPE) tissue samples and evaluate the relation to clinicopathological parameters, since it can be important to find an impact of each miR in PTC pathogenesis and lead the way for further studies to find a prognostic biomarker of PTC.

## 2. Materials and Methods

### 2.1. Human Tissue Samples

In this study, FFPE PTC tissues samples from 312 individuals were analyzed. The PTC tissues were obtained from patients who underwent total thyroidectomy at the Hospital of Lithuanian University of Health Sciences Kaunas Clinics between 2003 and 2017. After thyroidectomy postoperative radioactive iodine-131 (RAI) ablation was used for all patients; also, Levothyroxine doses that suppress serum TSH levels were prescribed. Lymphadenectomies were performed in 179 out of 312 patients. The patients were classified according to the eighth edition of the tumor–node–metastasis classification system (TNM-8th) [28]. N0a and N0b were assigned to category N0 according to TNM 8 (N0: No evidence of regional lymph node metastasis (*n* = 264); N0a: One or more cytologic or histologically confirmed benign lymph nodes (*n* = 179); N0b: No radiologic or clinical evidence of locoregional lymph node metastasis (*n* = 85)).

The study was approved by the Kaunas Regional Committee of Biomedical Research (Lithuania, approval No. BE-2-44; 2015-12-23). Written informed consent was obtained from each participant of the study after full explanation of the purpose and nature of all procedures used. This study was conducted in accordance with the Declaration of Helsinki.

### 2.2. microRNA Isolation and Reverse Transcription-Quantitative PCR

The miRNeasy FFPE Kit (Qiagen, Hilden, Germany) was used to isolate miRs from 5–10 mm^3^ sections of FFPE PTC tissues, following the manufacturer’s instructions. The PTC tissue samples for RNA extraction were macrodissected from areas that contained over >90% of malignant tissue. RNA quality and concentration were examined by NanoDrop 2000 Spectrophotometer (ThermoFisher Scientific, Waltham, MA USA). miRs were stored at −80 °C until further analysis.

miR expression was analyzed in triplicate via quantitative reverse transcription polymerase chain reaction (qRT-PCR) using a TaqMan Small RNA Assay (Applied Biosystems, USA) as we previously reported [23]. The expression of each miR was determined relative to that of let-7a and calculated by using the 2−ΔCq method. Relative fold-changes were estimated with the 2-ΔΔCq method [29].

### 2.3. Statistical Analysis

The normality of data distribution was tested using Kolmogorov–Smirnov criteria. The association between qualitative values in comparative groups was assessed by the Chi-square (χ2) test. FFPE tissue miR expression in relation to the clinicopathological PTC features was evaluated with the Mann–Whitney U test as the data distribution was not normal.

Patients with miR expression below the median values were assigned as having low expression levels, and patients with miR above or equal to median assigned as having high expression levels. The association between miR expression levels, clinicopathological features and overall survival (OS) were assessed by the Kaplan–Meier method. A log-rank test was used to estimate the statistical differences in Kaplan–Meier curves. OS was defined from the time of surgery to time of death or last follow-up. Survival time was censored for patients alive at the end of the study period. Univariate Cox proportional hazard regression model was performed to evaluate significant clinicopathological and molecular parameters for OS. Multivariate Cox proportional hazard regression model with enter method was conducted to evaluate independent prognostic predictors for OS.

All statistical analyses were performed using SPSS software (version 25.0, IBM, Armonk, NY, USA). A *p* value < 0.05 was considered as statistically significant.

### 2.4. The Cancer Genome Atlas (TCGA) Database Analysis

TCGA-Thyroid cancer (THCA) project data was used for the primary evaluation of miRs expression in tissue samples. The data groups were compared using t-test with Bonferroni correction (Statannot version 0.2.3; https://github.com/webermarcolivier/statannot) [30]). The survival data were evaluated with log-rank test (Lifelines version 0.25.7; https://zenodo.org/record/4313838#.YAf6L8WmO3J [31]).

## 3. Results

### 3.1. Study Population 

A total of 312 patients with papillary thyroid carcinoma (PTC) participated in the study. The demographic and clinicopathological characteristics of the study population are shown in Table 1.

### 3.2. Papillary Thyroid Carcinoma Tissue miR-146b, -21, -221, -222, -181b Expression in Relation with Clinicopathological Features

#### 3.2.1. TNM and miR Expression

PTC tumor sizes were distributed as follows: 166 (53.2%)-pT1a-pT1b, 146 (46.8%)–pT2-pT4. The majority of patients (121 out from 123) were at stage pT3 because the PTC of the extrathyroidal extension, 7 patients had a primary tumor ≥4 cm, 5 patients had PTC ≥4 cm and extrathyroidal extension of a tumor. PTC tissue miR-146b, miR-21, miR-221, miR-222, and miR-181b expression was compared depending on tumor size (pT1a, pT1b vs. pT2, pT3, pT4). In a group of PTC patients with pT1a and pT1b sized tumor, the expression of miR-146b, miR-21, miR-221, and miR-222 in PTC tissue samples was significantly lower than in patients with pT2, pT3, pT4 (*p* = 0.032; 0.0044; 0.003; and 0.001, respectively) (Figure 1, Table A1). Only miR-181b did not show significant difference between these groups (*p* = 0.317) (Figure 1, Table A1).

Lymph node metastases at initial surgery were observed in 48 (15.38%) of the PTC patients. Patients with lymph node metastases had significantly higher expression of miR-21, -221, -222 and -181b than those with no lymph node metastases (*n* = 264) (*p* < 0.05). Only miR-146b did not show significant difference between these groups (*p* = 0.057) (Figure 2, Table A1).

#### 3.2.2. Histological Features and miR Expression

Multifocality was observed in 68 (21.7%) of the PTC patients (Table 1). Patients with multifocal lesions had significantly lower expression of miR-21 than those with unifocal (*n* = 244) lesions (*p* < 0.011) in our study. miR-21 expression in multifocal PTC was 1.177 ± 0109 compared to single tumor 1.658 ± 0.096. (Figure 2; Table A1). miR-146, -221, -222, and -181b expression did not statistically significantly differ between these groups (Figure 3; Table A1).

Extrathyroidal extension was observed in 130 (41.5%) of the patients. miR-221, miR-222 expression was significantly higher in the PTC tissue samples with extrathyroidal extension than without (*p* = 0.049, *p* = 0.03, respectively) (Figure 4. Table A1). The expression of other investigated miRs did not differ significantly between these groups (Figure 4, Table A1).

PTC tissue samples of patients with autoimmune thyroiditis had statistically significantly higher expression of miR-21, than in those without autoimmune thyroiditis (*p* = 0.007) (Figure 5, Table A1). No significant difference was observed while comparing expression of miR-146b, -221, -222, and -181b in these groups (Figure 5, Table A1).

miR-146b, -21, -221, -222, and -181b expression in PTC lesions was compared in following groups: by age (<55 vs. ≥55 years), sex (male vs. female), presence of lymphovascular invasion (yes vs. no), however no statistically significant differences were found (Table A1).

Among all the patients, 88 (28.20%) had the classical variant of PTC, 36 (11.54%) had the follicular variant of PTC, 30 (9.62%) had the diffuse sclerosing variant of PTC, 105 (33.65%) had microcarcinoma, and 53 (16.99%) had oxiphylic cell carcinoma. We compared PTC tissue miR-146b, miR-21, miR-221, miR-222, and miR-181b expression in aggressive (diffuse sclerosing variant of PTC [32] and oxiphylic cell carcinoma [33,34]) histology variants of PTC to other, non-aggressive subtypes of PTC. Only miR-181b showed significantly lower tissue expression in aggressive histology variants of PTC compared to non-aggressive subtypes of PTC (*p* = 0.042) (Table 2).

### 3.3. Influence of Clinicopathological Features and miR Expression on OS.

The median follow-up time of PTC patients was 152 (IQR 60) months. Thirty-five deaths were recorded by the end of study. To evaluate PTC survival after thyroidectomy association with miR expression patients were divided into high and low PTC tissue miRs expression groups.

Patients with miR expression below the median values (miR-146b= 2.59; miR-21 = 1.16; miR-221 = 0.79; miR-222 = 2.04; miR-181b = 0.00048) were assigned as having low expression levels, and patients with miR above or equal to median assigned as having high expression levels. Kaplan–Meier curves estimating PTC patient OS after thyroidectomy according to high/low tissue expression levels of miR-146b, -21, -221, -222, and -181b curves were compared using the Log rank test, though no statistically significant differences were found (*p* = 0.479; *p* = 0.583; *p* = 0.383; *p* = 0.995; and *p* = 0.516) (Figure A1). Kaplan–Meier plots for miR-146b paired with miR-221 (median survival 141.7 months (IQR 18)) (A), and miR-146b, miR-221, miR-21 panel (median survival 138 months (IQR 12.75)) (B) is reported in Figure 6. The Log-rank test demonstrated significant differences in survival curves for miR-146b, miR-21 and miR-221 panel (*p* = 0.019). Higher expression of these 3 miRs panel is associated with decreased OS.

The median follow-up time in younger than 55 years group was 158 (IQR 59.25) months and median time ≥55 years group was 148 (IQR 65) months. The median follow-up time in males was 153 (IQR 63.75), in females 152 (IQR 60) months, in lymph node metastases positive group 150 (IQR 70) and negative group 153(IQR 60) months, in pT1a-pT1b 154 (IQR 59) and pT2-pT4 150 (IQR 63) months. Kaplan–Meier plots for age (< 55 vs. ≥55 years), sex (male vs. female), tumor size (pT1a- pT1b vs. pT2-pT4), lymph node metastases (negative vs. positive) are reported in Figure 7. The Log-rank test demonstrated significant differences in survival curves (*p* < 0.0001, *p* = 0.036, *p* = 0.021, *p* = 0.009, respectively). Ten-year survival was 97.7% in patients who had thyroidectomy younger than 55 years and 83.2% ≥55 years old and older, 95.5% in females and 89.07% in males, 94.34% in patients with pT1a and pT1b and 86.8% in patients with pT2-pT4, 80.9% in patients with present lymph node metastases and 93.2% in patients with absent lymph node metastases. The remaining clinicopathological parameters (autoimmune thyroiditis, lymphovascular invasion, extrathyroidal extension, multifocality) did not show significant differences (*p* > 0.05).

Univariate Cox regression hazards model analysis included clinicopathological features and the expression levels (high/low) of two different investigated miRs’ panels. The analysis revealed that females, younger age (<55 years) and patients with lower tumor size (pT1a-pT1b) are associated with longer OS. Presence of metastatic lymph nodes and high expression levels of miR-146b, miR-21, miR-221 were associated with increased hazard of shorter OS. After multivariate Cox proportional regression hazard model analysis, only sex (male) and age (equal or more than 55 years) emerged as independent prognostic factors associated with shorter OS (HR 0.28 (95% CI 0.09–0.86) and HR 0.05 (95% CI 0.01–0.22), respectively). (Table 3)

### 3.4. TCGA Database Analysis

#### 3.4.1. Selected miR Expression between Healthy and Cancerous Tissue

miR sequencing data from TCGA-THCA project were used to assess the expression levels. Additional filters were applied to represent our study population—patients diagnosed with papillary carcinoma, white race, not Hispanic or Latino ethnicity. In the TCGA analysis we used normal adjacent to the tumor (NAT) samples from PTC patients as a healthy sample group. We compared all PTC cancerous tissue samples with NAT samples (Figure A2). Then, we filtered out only those PTC cases which had paired cancerous and NAT samples and compered them as well (Figure A3).

#### 3.4.2. Selected miR Expression in Relation with Clinicopathological Features

TCGA-THCA miR expression analysis revealed no significant differences between sex groups (Figure A4). However, the higher expression of miR-146b, miR-221, miR-222, miR-181b were observed in a younger patient group. The age groups were divided into ≥55 years old and younger than 55 years old (Figure A5).

miR-146b, miR-221, miR-222 were upregulated in bigger and more extended (pT2, pT3, pT4) primary tumors than in smaller (pT1) tumors (Figure A6). Moreover, miR-146b, miR-21, miR-221, miR-222 overexpression was observed in a group with affected nearby lymph nodes (Figure A7).

Evaluating selected miRs’ ability to predict patient outcome only the expression of miR-181b was associated with patient OS (Figure A8). The higher than tumor median expression of miR-181b (log2RPM = 9.15) was linked with shorter OS of papillary carcinoma patients (*p* < 0.01).

## 4. Discussion

Proper clinicopathological risk stratification is important in the clinical management of PTC to improve the balance between treatment-associated benefits and risk of adverse complications [35]. Thorough assessment of clinicopathological features is considered useful in predicting clinical development [36]. Epigenetic biomarkers combined with clinicopathologic features of PTC might be beneficial in PTC patients risk stratification processes [10,32]. Five miRs (-146b, -21, -221, -222, and -181b) were selected for this study based on previous reports showing potential of these molecules in identifying and risk stratifying PTC patients [14,15,16,17,18,19,20,21,22,37]. According to TCGA analysis and previous studies, miR-146b, -21, -221, -222, -181b have higher expression levels in PTC compared to healthy thyroid tissue or nodular goiter [14,15,16,17,18,19,20,21,22,33]. In our study, we analyzed 312 PTC patients by exploring the expression levels of miR-146b, -21, -221, -222, -181b in FFPE tissue samples and evaluated relationship with clinicopathological parameters. The higher expression of miR-21 was related to unifocal lesions, autoimmune thyroiditis. miR-221, miR-222 expression was higher in the PTC tissue samples with extrathyroidal extension. In a group of PTC patients with pT1a and pT1b sized tumor, the expression of miR-146b, miR-21, miR-221, miR-222 in PTC tissue samples was lower than in patients with pT2, pT3, pT4. Patients with lymph node metastases had higher expression of miR-21, -221, -222, and -181b. The Log-rank test demonstrated significant differences in survival curves for miR-221, -21, -146b panel as higher levels of these miRs were associated with decreased OS. Age ≥ 55 years, male sex, lymph node metastases, pT2, pT3 and pT4 (TNM) are the clinicopathological parameters related to shorter OS in patients with PTC. Univariate analysis revealed that females, younger age (<55 years) and patients with lower tumor size (pT1a, pT1b) are associated with longer OS. Presence of metastatic lymph nodes and high expression levels of miR-146b, miR-221, miR-21 were associated with increased hazard of shorter OS. After multivariate analysis, only sex (male) and age (≥55 years) emerged as independent prognostic factors associated with shorter OS.

To reinforce our results, we performed a TCGA analysis to assess differences in 5 miR expression depending on clinicopathological parameters. Many of the results overlap. Both TCGA analysis and our results suggest that analysed miRs expression was independent of sex. Both results showed higher expression levels of miR-146b, -221, -222 in pT2-4 compared to pT1a-pT1b. Moreover, miR-21, -221, and -222 had higher expression in PTC patients with lymph node metastases in TCGA analysis and in our study results. Kaplan–Meier curves estimating PTC patients’ OS after thyroidectomy according to high/low tissue expression levels of miR-146b, -21, -221, and -222 curves were compared using the Log rank test, though no statistically significant differences were found in both analyses. However, there were also differences between the results of our study and the TCGA analysis. According to our results, the age of the patient has no effect on miR expression, however, TCGA analysis showed that higher expression of miR-146b, -221, -222, and -181b was characteristic of younger PTC patients. The results of our study also showed that the higher expression of miR-21 was also associated with higher T (TNM), whereas miR-181b expression was statistically significantly higher in PTC patients with lymph node metastases. TCGA analysis showed that higher miR-181b expression was associated with shorter OS, and in our study, we only obtained a significant association of higher expression of the mir-146b, -21, and -221 palette with shorter OS. As there are still some differences found and some questions left, our study results strengthened by TCGA analysis showed miR-146b, -21, -221, -222, and -181b relation with clinicopathological features of PTC.

Our study results revealed that a higher expression of miR-21 was related to autoimmune thyroiditis, lymph node metastases and pT2, pT3, pT4 (TNM) compared to pT1a, pT1b. Huang et al. also found miR-21 overexpression relation to lymph node metastases [38]. Zhang et al revealed miR-21 overexpression relation to extrathyroidal extension, lymph node metastases, advanced TNM stage (III/IV) [15]. MiR-21 promotes cell proliferation and invasion via the VHL/PI3K/AKT pathway [39], and probably its expression increases as tumor grows or spreads.

Multifocality empirically is often treated as a risk factor for aggressive course of PTC, encouraging aggressive treatments [40]. However, inconsistency and even contradiction in a literature is present concerning the role of tumor multifocality in clinical outcomes of PTC [41,42,43,44,45,46,47]. Therefore, the prognostic value of multifocality of PTC remains controversial, creating a difficulty in the current clinical management. Our results suggest that lower expression of miR-21 is related to multifocal lesions (*p* < 0.011). miR-21 expression in PTC tissue is inconsistent compared to healthy tissue in a literature. [17,19,48]. However, miR-21 was overexpressed in PTC with aggressiveness linked to clinicopathological features in FFPE PTC tissue samples of patients in our study. miR-21 is associated with aggressive behaviors and poor survival in some other cancers as well [49,50,51,52]. Our findings might be a biomolecular proof that-multifocality is not a poor PTC prognosis risk factor. However, Zhang et al. found that miR-21 levels were significantly higher in patients with multifocal lesions (*p* < 0.005) [53]. Molecular multifocal independent primary PTC separation from intrathyroid metastasis of PTC might be important for predicting the lymph node metastasis, aggressiveness, and prognosis of PTC [54]. Moreover, this might be a key factor explaining miR-21 expression differences (possible overexpression in PTC with intrathyroidal metastastases, but lower expression in multifocal primary PTC). Further studies investigating the impact of multifocality to PTC prognosis as well as relation with miR expression is needed.

Most studies have shown a significant relationship between thyroid cancer and positive antibodies to thyroglobulin and histological proof of AT [55]. Since AT and PTC share some risk factors: greater incidence in women, and in patients after radiotherapy of the neck, iodine deficiency [55]. Both disorders have genetic link: RET/PTC rearrangements could be more often found in carcinomas associated with AT, but this mutation could be found in only AT, as well [56]. As we found miR-21 overexpression in PTC patients with AT compared to patients without this disease (*p* < 0.007), this molecule might play an important role in a pathogenesis of both diseases as well.

Previous studies have revealed that larger tumor size was associated with increased incidence of nodal spread and worse prognosis of PTC [57,58]. Several reports have also suggested tumor size as a predictor for central lymph node metastases in PTC patients [59]. We found that miR-146b, miR-21, miR-221 and miR-222 tend to be overexpressed in pT2, pT3, pT4 (TNM) compared to pT1a, pT1b. TCGA analysis revealed miR-146b, miR-222 and miR-221 to be overexpressed in pT2, pT3, pT4 vs pT1a, pT1b. miR-146b, miR-222 had also higher expression levels in higher PTC stages in studies by Zhang et al. [53] and Wang et al [26]. miR-221 was also previously found to be overexpressed in higher TNM stages [17,26].

Lymph node metastases in PTC has been shown to be associated with an increased risk of regional recurrence, poor prognosis and decreased survival, especially in older patients. Hence, there is a need for a reliable biomarker for the prediction of lymph node metastases in this cancer [60]. In our study, patients with lymph node metastases had higher expression of miR-21, -221, -222, and -181b (*p* < 0.05). The tendency of higher miR-146b expression in PTC with lymph node metastases was also obvious, thus we did not get significance (*p* = 0.057). TCGA analysis also showed significantly higher PTC tissue expression of miR-146b, -21, -221, -222 in patients with lymph node metastases, though miR-181 was equally expressed in both groups. Other studies comparing PTC patients with and without lymph node metastases showed upregulated miRs (miR-146, miR-222, miR-221, and miR-146a) in patients with present lymph node metastases [15,18,25,28]. miR-146b, -21, -221, -222, and -181b expression might be useful tool in risk stratifying and selecting radicality of treatment (need of iodine-131I ablation) as well as intensity of follow up plan for PTC patients.

The high expression of four miRs (miR-146a, miR-146b, miR-182, and miR-203) and the low expression of six miRs (miR-1271, miR-791, miR-381, miR-let 7a, miR-26a, miR-486) was correlated with decreased OS in PTC [61]. We did not get significant correlations of miR-222, -221, -146b, -181b, -21 with OS in our study. However, the Log-rank test demonstrated significant differences in survival curves for miR-146b, miR-221 and miR-21 panel (*p* = 0.019). Higher expression of these 3 miRs panel was associated with decreased OS. Interestingly, overexpression of miR-146b [61], miR-21 [23] were found to be associated with shorter disease-free survival in PTC patients. Since these miRs have interfaces with disease free survival and OS their expression may be important in predicting the course of PTC.

Diffuse sclerosing variant of PTC and oxiphylic cell carcinoma are being reported as variants of PTC associated with aggressive behaviors and poor prognosis [32,33,62,63,64]. We had 30 patients with diffuse sclerosing variant and 53 with oxiphylic cell carcinoma in our study (83 samples of agrresive histology PTC). We found that expression of miR-181b tends to be lower in aggressive variants of PTC. However, Li et al. showed that downregulation of miR-181b expression causes cellular growth inhibition, promoting cellular apoptosis by targeting cylindromatosis gene in papillary thyroid cancer [65]. Futher studies of miR-181b role in different PTC histological variants pathogenesis is needed as we resulted showing that PTC aggressiveness is associated with lover levels of miR-181b.

Male sex and older age at a time of diagnosis of PTC correlated with decreased OS as it is expected by epidemiology. Larger tumor size and lymph node metastases correlated with shorter OS. Unless univariate analysis showed increased hazard in OS in PTC patients with higher levels of 3 miRs (miR-146b, 221, -21) panel, lymph node metastasis presence, and older males, after multivariate analysis only sex (male) and age (≥ 55 years) emerged as independent prognostic factors associated with shorter OS in PTC patients. To our best knowledge in previous miR expression in PTC tissue studies, only lower expression of miR-26a [66] and higher expression of miR-182 [67] had poorer OS rates and after multivariate analysis were showed to be independent prognostic risk factors of decreased OS in PTC.

The strength of our study is quite big sample size. We investigated 312 PTC tissue samples and concentrated our attention to miR relation to clinicopatologic features. Moreover, our results about 5 miR expressions’ relation to clinicopathologic PTC parameters largely overlapped and coincided with the TCGA analysis data. The limitation of our study was that we analyzed only 5 miR expression in PTC tissue samples since other miR molecules (for example miR-146a, miR-182, miR-203, miR-1271, miR-791, miR-381, miR-let 7a, miR-26a, and miR-486 [61]) might also have an important relations and possible value in risk stratification of PTC patients as well.

In conclusion, miRs associations with high-risk PTC features like larger tumor size (overexpression of miR-146b, miR-221, miR-222) and lymph node metastases (overexpression of miR-21, -221, -222, -181b) and higher expression of miR-146b, -21, and -222 panel association with decreased OS show that these 5 miRs have possible value for risk stratification and prognosis of PTC and should be further evaluated. The relation of higher expression of miR-21 with autoimmune thyroiditis and unifocal PTC compared to multifocal shows the need of further investigations of miR-21 role in pathogenesis of PTC.

## Figures and Tables

**Figure 1 diagnostics-11-00418-f001:**
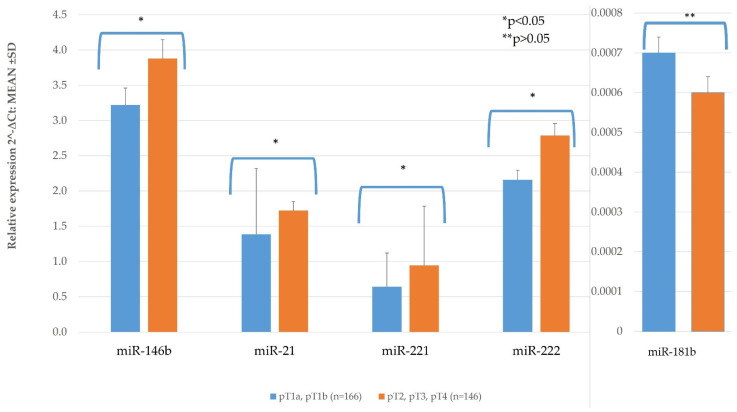
The comparison of papillary thyroid carcinoma (PTC) tissue miR expression depending on tumor size (pT1a, pT1b vs. pT2, pT3, pT4). All data are presented as the mean ± SD. * *p* < 0.05, ** *p* > 0.05.

**Figure 2 diagnostics-11-00418-f002:**
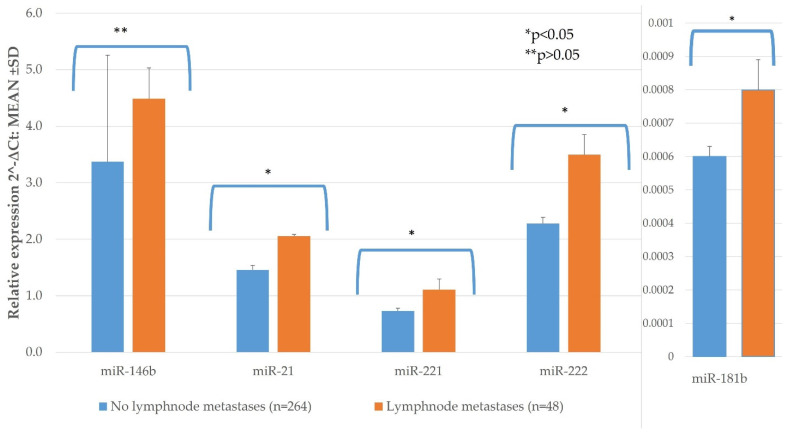
The comparison of PTC tissue miR expression between lesions with lymph node metastases (*n* = 48) and without (*n* = 264). All data are presented as the mean ± SD. * *p* < 0.05, ** *p* > 0.05.

**Figure 3 diagnostics-11-00418-f003:**
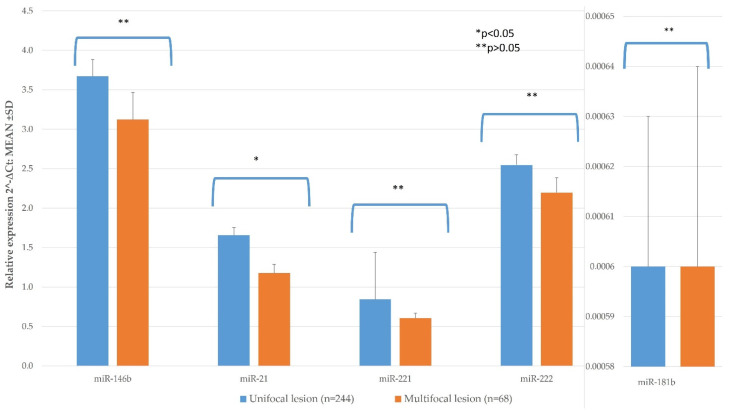
The comparison of PTC tissue miR expression between multifocal and unifocal lesions. All data are presented as the mean ± SD. * *p* < 0.05, ** *p* > 0.05.

**Figure 4 diagnostics-11-00418-f004:**
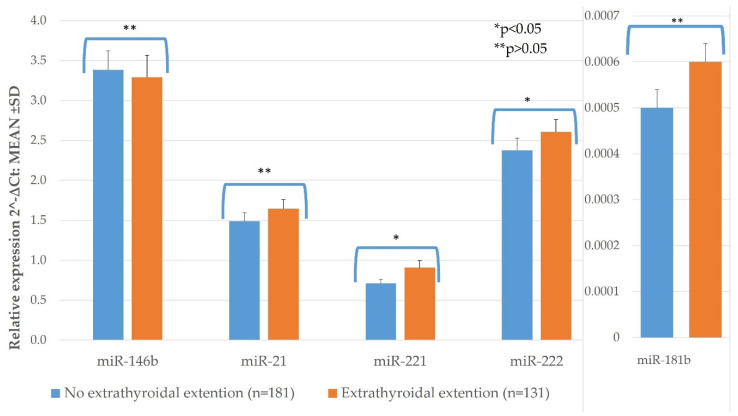
The comparison of PTC tissue miR expression between lesions with extrathyroidal extension (*n* = 131) and without (*n* = 181). All data are presented as the mean ± SD. * *p* < 0.05, ** *p* > 0.05.

**Figure 5 diagnostics-11-00418-f005:**
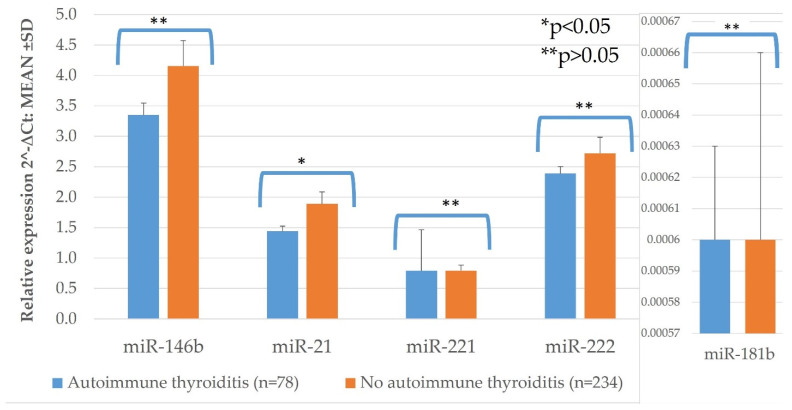
The comparison of PTC tissue miR expression between lesions in patients with autoimmune thyroiditis (*n* = 78) and without (*n* = 234). All data are presented as the mean ± SD. * *p* < 0.05, ** *p* > 0.05.

**Figure 6 diagnostics-11-00418-f006:**
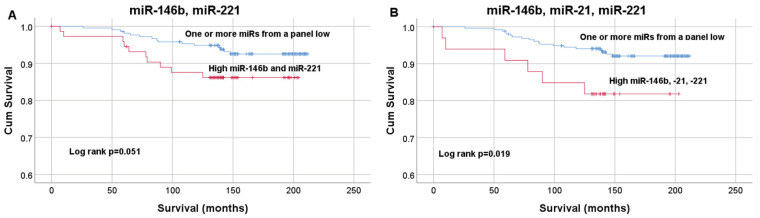
Kaplan–Meier curves estimating PTC patient survival after thyroidectomy according to high/low tissue expression of miR-146b and miR-221 (**A**), miR-146b, miR-21, miR-221 panel (**B**). OS curves were compared using the Log rank test.

**Figure 7 diagnostics-11-00418-f007:**
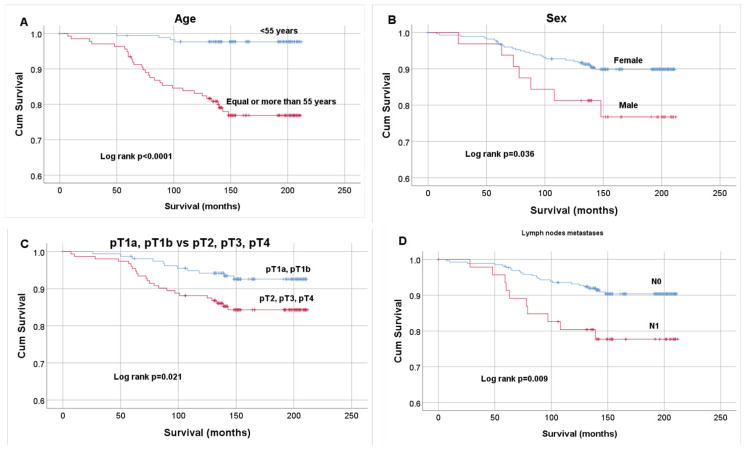
Kaplan–Meier curves estimating PTC patient survival after thyroidectomy according to age (**A**), sex (**B**), T (TNM) (**C**), lymph node metastases (N0- absent, N1- present) (**D**). Overall survival (OS) curves were compared using the Log rank test.

**Table 1 diagnostics-11-00418-t001:** Demographic and clinicopathological characteristics of the study population.

Characteristic	All Patients	*p*
	(*n* = 312)	
Sex, *n* (%)		*p* < 0.05
Male	33 (10.5%)	
Female	279 (89.1%)	
Age at initial surgery (years)	53 (min = 18, max = 83)	*p* < 0.05
Median (min-max)	174 (55.77%)	
<55 years, *n* (%)	138 (44.23%)	
≥55 years, *n* (%)		
Tumor size, *n* (%)		*p* < 0.05
pT1a	100 (32.05%)	
pT1b	66 (21.15%)	
pT2	18 (5.78%)	
pT3	123 (39.42%)	
pT4	5 (1.6%)	
Lymph node metastases at initial surgery,	264(84.7%)/48(15.3%)	*p* < 0.05
*n* (%)		
no/yes		
Variant of PTC, *n* (%)		*p* < 0.05
The classical variant	88 (28.20%)	
The diffuse sclerosing variant	30 (9.62%)	
Microcarcinoma	105 (33.65%)	
Oxiphylic cell carcinoma	53 (16.99%)	
The follicular variant	36(11.54%)	
Multifocality,	244 (78.2%)/68 (21.8%)	*p* < 0.05
*n* (%)		
no/yes		
Extrathyroidal extension,	182 (58.4%)/130 (41.6)	*p* < 0.05
*n* (%)		
no/yes		
Lymphovascular invasion,	139 (44.6%)/173 (55.4%)	*p* > 0.05
*n* (%)		
no/yes		
Autoimmune thyroiditis,	234 (75%)/78 (25%)	*p* < 0.05
*n* (%)		
no/yes		

**Table 2 diagnostics-11-00418-t002:** PTC tissue miR levels in aggressive histology variants of PTC compared to non-aggressive subtypes of PTC.

miR	Relative Expression 2^-∆Ct^: MEAN ± SD		*p*-Value
	Aggressive Histology of PTC (*n* = 83)	Non-Aggressive Subtypes of PTC (*n* = 229)	
146b	3.54 ± 2.83	3.55 ± 3.23	0.741
21	1.52 ± 1.39	1.55 ± 1.38	0.772
221	0.81 ± 0.66	0.79 ± 0.88	0.327
222	2.64 ± 1.94	2.45 ± 1.93	0.365
181b	0.0004 ± 0.0002	0.0006 ± 0.0005	0.042

**Table 3 diagnostics-11-00418-t003:** Univariate and multivariate analysis of OS in patients with PTC. COX proportional regression hazard model was used.

	Univariate Analysis		Multivariate Analysis	
	HR (95% CI)	*p*-Value	HR (95% CI)	*p*-Value
Sex *(female vs. male)*	0.42 (0.18–0.96)	0.042	0.28 (0.09–0.86)	0.026
Age *(<55 y vs. ≥55 y)*	0.1 (0.03–0.27)	<0.001	0.05(0.01–0.22)	<0.001
Tumor size *(pT1a-pT1b vs. pT2-4)*	0.44 (0.21–0.9)	0.024	0.84 (0.35–2.01)	0.69
Lymph nodes (*pos vs. neg*)	2.59 (1.24–5.42)	0.011	2.16 (0.73–6.35)	0.162
Multifocality *(yes/no)*	1.49 (0.71–3.12)	0.29	1.37 (0.52–3.61)	0.527
Extrathyroidal extension *(yes/no)*	1.77 (0.9–3.49)	0.098	1.43 (0.41–4.95)	0.57
Lymphovascular invasion *(yes/no)*	1.77 (0.86–3.63)	0.121	1.45 (0.62–3.39)	0.393
Aggressive histology *(yes/no)*	0.93 (0.29–3.04)	0.904	0.96 (0.19–4.98)	0.962
miR-146b & miR-221 *(high vs. low)*	2.17 (0.98–4.85)	0.057	1.77 (0.53–5.89)	0.354
miR-146b & miR-221 & miR-21 *(high vs. low)*	2.86 (1.14–7.19)	0.026	1.37 (0.37–5.08)	0.64

## Data Availability

The results published here are in part based upon data generated by the TCGA Research Network: https://www.cancer.gov/tcga.

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
