# Peer review of "Papillary Thyroid Carcinoma Tissue miR-146b, -21, -221, -222, -181b Expression in Relation with Clinicopathological Features"

_diagnostics, 2021, doi:10.3390/diagnostics11030418_

Round 1

Reviewer 1 Report

The Authors undertaken the analysis of the usefulness of some potential biomarkers in diagnostics and treatment of the papillary thyroid cancer. The work is performed quite well, design is well tailored, however there are some fundamental, substantive inaccuracies (or we can even say mistakes), which have to be reconsidered or (we can even say) corrected before further evaluation. I would like to encourage the authors for the thorough analysis and revision. Please see the remarks listed below.

  1. What was the decision about the order of the presentation of the analyzed miRNA fragments. In the title you started with 222, 221, next 21? etc, in the abstract with 221 next 222, and next 146b? - please reconsider it, however I would recommend presenting the analyzed miRNAs in one established order throughout the whole manuscript (especially in the title and abstract).
  2. Why sometimes you use abbreviation "MiRNA" using capital letter and sometimes "miR"? sometimes “mir” or “MiR”. Please decide to use one proper form. I understand that maybe a lot of abbreviations are used in the articles all over the world, however I would recommend to use one standard abbreviation (maximum two). It would be better i.e. more understandable for potential readers and it would look smart. In this form it looks like a messiness.
  3. When you use some abbreviation for the first time, even if it is very commonly used, please for the first time use the whole name (also in the abstract, sometimes readers read the abstract, and after it they decide about farther reading of the whole article, so it must be clear and understandable).
  4. It were the pT1a and pT1b tumors?, i.e. after histopathological analysis?, if yes use the letter “p” before the number indicating his size (in the whole manuscript). The same with T2, T3, T4.
  5. “FFPE” – when you use it for the first time in the manuscript, please use the whole name before it (also in the abstract).
  6. When you use the term “overall survival”, please after it, put in the abbreviation “OS”, do it only one, and after it, use only abbreviation of it throughout the whole manuscript.
  7. When you relate to the high frequency of the incidence of PTC please use some more up to date references not from 2012, or even 2015 (we have 2021).
  8. The references are written very chaotic, without the journal's guidelines. Please check the guidelines and introduce them.
  9. One of the prognostic risk factor for PTC is the age of 55 years old and older, not only above 55. Please correct it in the text.
  10. Please do not use in scientific analysis the term “etc.” what does it mean?, what do you mean by it. It may mean, that the characteristics, which you enumerated were not strict, accurate and not important, that anyone may end this sentence using whatever features.
  11. The features which you enumerate like thyroid extension, lymph node metastasis, lymphovascular invasion are not clinical, are rather pathological, which we obtain after surgery.
  12. The statement “Recently, an attention has shifted to revealing bio molecular markers of thyroid cancer. The European Thyroid Association advises investigating BRAF, RET/PTC,PAX8/PPARG, RAS mutations, TERT promoter, and TP53 in case of hesitation about further clinical action depending on Bethesda classes, but these biomarkers are not recommended for follow-up yet.” is not clear. You said, that after Bethesda diagnosis some other molecular tests are taken under consideration, however these molecular test are performed in histopathological specimens, not cellular, so after surgery, not after diagnosis established according to Bethesda classification. Please clarify.
  1. After thyroidectomy postoperative radioactive iodine-131 ablation was used for all patients? Also in patients with microcarcinoma? All patients before surgery had estimated diagnosis before surgery. All patients had radical surgery performed with lymphadenectomy? Do you have any information about it. It would be well to have it in materials and methods section.
  2. Why the references used in the manuscript are not numerated according to the order of their appearance in the text? Please check the journal’s guidelines and correct them according to these recommendations.
  3. This is not clear what tissues were examined: 312 tissues of PTC? 312 tissues of the thyroid glands with PTC? How many individuals were analyzed de facto? 312? Please introduce more accurate and clear description especially in the abstract and materials and methods section, the information included in the text are misleading.
  4. Please decide to use the term “tumor” or “tumour” and use one term in the manuscript. Each of them are characteristic for one accurate language type, and should be use in the one, accurate type of language.
  5. The quality of the figures are low, I could not recognize some details. Before final publication please increase their resolution.
  6. Table 1: what was the age of the patients at initial surgery?
  7. Why so many patients were at stage pT3? The tumors were larger than 4 cm across, or they has just begun to grow into nearby tissues outside the thyroid? Please clarify it in the text, it is extremely important for the analyzed population description.
  8. Table 1 “Microcarcinoma” is not the histological variant of PTC, the term “microcarcinoma” tells us only the information about the size of PTC not about the histology. The statistical analysis has to be reanalyzed.
  9. What the asterisk beside the term Tumor size means?, where is the description?
  10. In table 1 you wrote that tumors T1 were 166 and T2+T3 were 141 but in the figure 8 you wrote the number T2, T3 n=152 why?, you did perform analysis more than one of the same samples?
  11. You wrote that lymph node metastasis had 48 patients, multifocality was observed in 68, ETE had 173 patients, lymphovascular invasion had 173 patients, and in the further analysis you divided the PTC for aggressive and non-aggressive entities (83 and 229) so on basis which parameters did you perform this division? Please clarify, generally oxiphylic cell carcinoma is not assigned to aggressive form.
  12. In the sentence “After multivariate Cox proportional regression hazard model analysis, only gender (male) and age (55 years) emerged … “ rather should be “equal or more than 55”, not only “55.”
  13. Decide to use the term “sex” or “Gender”, however, using the term “sex” sounds better in the meaning of not social but medical meanings. I recommend it.
  14. You cited, that: “Zhang et al revealed miR-21 overexpression relation to extrathyroidal extension, lymph node metastases, TNM stage.” what TNM stage (higher? lower?), please clarify.
  15. You wrote that: “MiRNA-21 promotes cell proliferation and invasion via the VHL/PI3K/AKT pathway, and probably its expression increases as tumor grows or spreads.” however in your study you have estimated, that higher expression of miRNA-21 was related to unifocal lesions. It should be discussed in the discussion section.
  16. “Multifocality empirically is often treated as a risk factor for PTC” this sentence is not clear, multifocality is not treated as a risk factor for PTC, but rather as a risk factor of aggressive course of PTC.
  17. It is not clearly said, if you analyzed PTC tissue and compared it with healthy (not malignant) ones? If yes, from which patients they were obtained? From the same patients? It should to be clearly written.
  18. The authors have to declare what is the main goal of their study because the opinion that “Combination of neck ultrasound and these 5 miRNAs expression might be useful risk stratifying and selecting radicality of treatment of PTC.” is ambiguous. Ultrasound is the tool for pre operation analysis and is of course useful for treatment planning (the range of surgery for example total thyroidectomy, + lymphadenectomy), but miRNA expression level we obtain after surgery (post factum, after surgical treatment) so, what about the significance of these parameter.   
  1. Again, the authors use the abbreviation like “CYLD” and they do not present the whole name of it. They even do not present it in Abbreviation section. It looks like the authors do not understand what they are writing about.
  2. “(…) older age at a thyroidectomy correlated with decreased OS as it is expected by epidemiology.” We do not have the evidence regarding the age of the patient, when she/he undergo operation and OS. We rather have the evidence regarding the age of the patients at the time of diagnosis (with surgical treatment) of PTC and their OS. Please clarify.
  3. Some spelling errors like “ir” and some others should be eliminated before further proceedings.
  4. The conclusions should not be the repetition of the results, but should be some thoughts revealing from the analysis.
  5. The references are written completely without any rules and standards and should be re-enumerated and re-written.

Author Response

Dear Reviewer,

enclosed with this letter is the revised version of our manuscript “Papillary thyroid carcinoma tissue miR-146b, -21, -221, -222, -181b expression in relation with clinicopathological features “, which we would like to resubmit for publication as a research article in Diagnostics.

We have taken up your criticism and suggestions. In the accompanying rebuttal letter are our point-by-point responses to each comment. In the revised manuscript, all changes have been indicated by “Track changes” function or highlighted.

We expect that the revisions in the manuscript along with the accompanying responses will be considered for publication.

Birute Zilaitiene

Reviewer 2 Report

The data presented in the article are not completely new. The expression of MiRNA-222, -221, -21, -146b, -181b has been described in a fairly large number of works. The article itself makes an ambiguous impression. The sample is representative, the study design is well thought out. But. questions arise. 1. Why use the Cancer Genome Atlas (TCGA) database data if they practically do not differ from the results obtained by the authors? In my opinion, this data from Figure 1 to Figure 7 can be safely transferred to additional materials in order not to overload the article. 2. The description of the results should definitely start with the description of the population. 3. The expression of all 5 miRNAs has not been compared by multivariate methods, for example, PCA, in order to establish which parameter has the greatest influence? It would be appropriate. 4. If the authors write that the distribution differs from the normal one, why is the confidence interval given in the tables? In Table 2, it is unlikely that siRNA-181b can differ significantly. Is there no typo? 5. When analyzing the survival rate, for some reason, the values ​​for individual miRNAs are not given, only for combinations. I wanted to see them separately, and also to understand why these combinations were chosen. Why aren't numerical values ​​of expression used, but only relative (low, high)? 6. In this study, the level of thyroid hormones was not determined? Have you not checked whether there is a connection between hormone levels and miRNA expression? I believe that the article needs substantial revision. 

Author Response

(The authors gave the same response as above.)

Round 2

Reviewer 1 Report

The authors revised the manuscript, almost all inaccuracies are corrected. However there is still one crucial remark, which should be discussed before further evaluation. The authors write about some diagnosed in their population study histological variants of PTC. They enumerate the classical variant of PTC (88), the diffuse sclerosing variant of PTC (30), oxyphylic cell variant (53), the follicular variant (36) and "microcarcinoma". THIS IS NOT THE HISTOLOGICAL VARINAT OF PTC, IT ONLY MEANS THAT THIS PTC TUMOR IS EQUAL OR BELOW 1.0CM OF DIAMETER, AND NOTHING MORE. Because papillary microcarcinomas are classified based on size (=<1 cm), these tumors do not exhibit a distinctive morphology. Rather, they can have features of any larger papillary carcinoma subtype. So I propose to change the title in the table 1 from "Histological variant of PTC" to "Variant of PTC" or "Subtype of PTC". 

Author Response

Manuscript ID: diagnostics-1108722

Papillary thyroid carcinoma tissue miR-146b, -21, -221, -222, -181b expression in relation with clinicopathological features

Dear Reviewer,

enclosed with this letter is the revised version of our manuscript “Papillary thyroid carcinoma tissue miR-146b, -21, -221, -222, -181b expression in relation with clinicopathological features “, which we would like to resubmit for publication as a research article in Diagnostics.

We have taken up your criticism and suggestions. In the accompanying rebuttal letter are our point-by-point responses to each comment. In the revised manuscript, all changes have been indicated by “Track changes” function.

We expect that the revisions in the manuscript along with the accompanying responses will be considered for publication.

Birute Zilaitiene

Responses to Reviewer 1 comments (Round 2):

The authors revised the manuscript, almost all inaccuracies are corrected. However there is still one crucial remark, which should be discussed before further evaluation. The authors write about some diagnosed in their population study histological variants of PTC. They enumerate the classical variant of PTC (88), the diffuse sclerosing variant of PTC (30), oxyphylic cell variant (53), the follicular variant (36) and "microcarcinoma". THIS IS NOT THE HISTOLOGICAL VARINAT OF PTC, IT ONLY MEANS THAT THIS PTC TUMOR IS EQUAL OR BELOW 1.0CM OF DIAMETER, AND NOTHING MORE. Because papillary microcarcinomas are classified based on size (=<1 cm), these tumors do not exhibit a distinctive morphology. Rather, they can have features of any larger papillary carcinoma subtype. So I propose to change the title in the table 1 from "Histological variant of PTC" to "Variant of PTC" or "Subtype of PTC". 

Response.: The title in the Table1 was changed from “Histological variant of PTC to “Variant of PTC”.

Spelling was checked once again and some inaccuracies corrected.

Reviewer 2 Report

The authors made significant changes to the text of the manuscript in accordance with the comments of the reviewers. The only remark / suggestion to the authors of the article is to change the scale in the figures with the Kaplan-Meier curves, for example, in the range 0.6 - 1. This will make the curves look better.

Author Response

Manuscript ID: diagnostics-1108722

Papillary thyroid carcinoma tissue miR-146b, -21, -221, -222, -181b expression in relation with clinicopathological features

Dear Reviewer,

enclosed with this letter is the revised version of our manuscript “Papillary thyroid carcinoma tissue miR-146b, -21, -221, -222, -181b expression in relation with clinicopathological features “, which we would like to resubmit for publication as a research article in Diagnostics.

We have taken up your criticism and suggestions. In the accompanying rebuttal letter are our point-by-point responses to each comment. In the revised manuscript, all changes have been indicated by “Track changes” function.

We expect that the revisions in the manuscript along with the accompanying responses will be considered for publication.

Birute Zilaitiene

Responses to Reviewer 2 comments:

Comment: The authors made significant changes to the text of the manuscript in accordance with the comments of the reviewers. The only remark / suggestion to the authors of the article is to change the scale in the figures with the Kaplan-Meier curves, for example, in the range 0.6 - 1. This will make the curves look better.

Response.: We corrected Kaplan- Meier curves as suggested (You can see them in a letter below and in corrected manuscript as well).
